# Streaming Variational Bayes

**Tamara Broderick,    Nicholas Boyd,    Andre Wibisono,    Ashia C. Wilson**
University of California, Berkeley
{tab@stat, nickboyd@eecs, wibisono@eecs, ashia@stat}.berkeley.edu

**Michael I. Jordan**
University of California, Berkeley
jordan@cs.berkeley.edu

## Abstract

We present SDA-Bayes, a framework for (S)treaming, (D)istributed, (A)synchronous computation of a Bayesian posterior. The framework makes streaming updates to the estimated posterior according to a user-specified approximation batch primitive. We demonstrate the usefulness of our framework, with variational Bayes (VB) as the primitive, by fitting the latent Dirichlet allocation model to two large-scale document collections. We demonstrate the advantages of our algorithm over stochastic variational inference (SVI) by comparing the two after a single pass through a known amount of data—a case where SVI may be applied—and in the streaming setting, where SVI does not apply.

## 1   Introduction

Large, streaming data sets are increasingly the norm in science and technology. Simple descriptive statistics can often be readily computed with a constant number of operations for each data point in the streaming setting, without the need to revisit past data or have advance knowledge of future data. But these time and memory restrictions are not generally available for the complex, hierarchical models that practitioners often have in mind when they collect large data sets. Significant progress on scalable learning procedures has been made in recent years [e.g., 1, 2]. But the underlying models remain simple, and the inferential framework is generally non-Bayesian. The advantages of the Bayesian paradigm (e.g., hierarchical modeling, coherent treatment of uncertainty) currently seem out of reach in the Big Data setting.

An exception to this statement is provided by [3–5], who have shown that a class of approximation methods known as *variational Bayes* (VB) [6] can be usefully deployed for large-scale data sets. They have applied their approach, referred to as *stochastic variational inference* (SVI), to the domain of topic modeling of document collections, an area with a major need for scalable inference algorithms. VB traditionally uses the variational lower bound on the marginal likelihood as an objective function, and the idea of SVI is to apply a variant of stochastic gradient descent to this objective. Notably, this objective is based on the conceptual existence of a full data set involving $D$ data points (i.e., documents in the topic model setting), for a fixed value of $D$. Although the stochastic gradient is computed for a single, small subset of data points (documents) at a time, the posterior being targeted is a posterior for $D$ data points. This value of $D$ must be specified in advance and is used by the algorithm at each step. Posteriors for $D'$ data points, for $D' \neq D$, are not obtained as part of the analysis.

We view this lack of a link between the number of documents that have been processed thus far and the posterior that is being targeted as undesirable in many settings involving streaming data. In this paper we aim at an approximate Bayesian inference algorithm that is scalable like SVI but

is also truly a streaming procedure, in that it yields an approximate posterior for each processed collection of $D'$ data points—and not just a pre-specified "final" number of data points $D$. To that end, we return to the classical perspective of Bayesian updating, where the recursive application of Bayes theorem provides a sequence of posteriors, not a sequence of approximations to a fixed posterior. To this classical recursive perspective we bring the VB framework; our updates need not be exact Bayesian updates but rather may be approximations such as VB. This approach is similar in spirit to assumed density filtering or expectation propagation [7–9], but each step of those methods involves a moment-matching step that can be computationally costly for models such as topic models. We are able to avoid the moment-matching step via the use of VB. We also note other related work in this general vein: MCMC approximations have been explored by [10], and VB or VB-like approximations have also been explored by [11, 12].

Although the empirical success of SVI is the main motivation for our work, we are also motivated by recent developments in computer architectures, which permit distributed and asynchronous computations in addition to streaming computations. As we will show, a streaming VB algorithm naturally lends itself to distributed and asynchronous implementations.

## 2   Streaming, distributed, asynchronous Bayesian updating

**Streaming Bayesian updating.** Consider data $x_1, x_2, \ldots$ generated iid according to a distribution $p(x \mid \Theta)$ given parameter(s) $\Theta$. Assume that a prior $p(\Theta)$ has also been specified. Then Bayes theorem gives us the *posterior distribution* of $\Theta$ given a collection of $S$ data points, $C_1 := (x_1, \ldots, x_S)$:

$$p(\Theta \mid C_1) = p(C_1)^{-1} \, p(C_1 \mid \Theta) \, p(\Theta),$$

where $p(C_1 \mid \Theta) = p(x_1, \ldots, x_S \mid \Theta) = \prod_{s=1}^{S} p(x_s \mid \Theta)$.

Suppose we have seen and processed $b-1$ collections, sometimes called *minibatches*, of data. Given the posterior $p(\Theta \mid C_1, \ldots, C_{b-1})$, we can calculate the posterior after the $b$th minibatch:

$$p(\Theta \mid C_1, \ldots, C_b) \propto p(C_b \mid \Theta) \, p(\Theta \mid C_1, \ldots, C_{b-1}). \tag{1}$$

That is, we treat the posterior after $b - 1$ minibatches as the new prior for the incoming data points. If we can save the posterior from $b - 1$ minibatches and calculate the normalizing constant for the $b$th posterior, repeated application of Eq. (1) is streaming; it automatically gives us the new posterior without needing to revisit old data points.

In complex models, it is often infeasible to calculate the posterior exactly, and an approximation must be used. Suppose that, given a prior $p(\Theta)$ and data minibatch $C$, we have an approximation algorithm $\mathcal{A}$ that calculates an approximate posterior $q$: $q(\Theta) = \mathcal{A}(C, p(\Theta))$. Then, setting $q_0(\Theta) = p(\Theta)$, one way to recursively calculate an approximation to the posterior is

$$p(\Theta \mid C_1, \ldots, C_b) \approx q_b(\Theta) = \mathcal{A}\left(C_b, q_{b-1}(\Theta)\right). \tag{2}$$

When $\mathcal{A}$ yields the posterior from Bayes theorem, this calculation is exact. This approach already differs from that of [3–5], which we will see (Sec. 3.2) directly approximates $p(\Theta \mid C_1, \ldots, C_B)$ for fixed $B$ without making intermediate approximations for $b$ strictly between $1$ and $B$.

**Distributed Bayesian updating.** The sequential updates in Eq. (2) handle streaming data in theory, but in practice, the $\mathcal{A}$ calculation might take longer than the time interval between minibatch arrivals or simply take longer than desired. Parallelizing computations increases algorithm throughput. And posterior calculations need not be sequential. Indeed, Bayes theorem yields

$$p(\Theta \mid C_1, \ldots, C_B) \propto \left[\prod_{b=1}^{B} p(C_b \mid \Theta)\right] p(\Theta) \propto \left[\prod_{b=1}^{B} p(\Theta \mid C_b) \, p(\Theta)^{-1}\right] p(\Theta). \tag{3}$$

That is, we can calculate the individual minibatch posteriors $p(\Theta \mid C_b)$, perhaps in parallel, and then combine them to find the full posterior $p(\Theta \mid C_1, \ldots, C_B)$.

Given an approximating algorithm $\mathcal{A}$ as above, the corresponding approximate update would be

$$p(\Theta \mid C_1, \ldots, C_B) \approx q(\Theta) \propto \left[\prod_{b=1}^{B} \mathcal{A}(C_b, p(\Theta)) \, p(\Theta)^{-1}\right] p(\Theta), \tag{4}$$

for some approximating distribution $q$, provided the normalizing constant for the right-hand side of Eq. (4) can be computed.

Variational inference methods are generally based on exponential family representations [6], and we will make that assumption here. In particular, we suppose $p(\Theta) \propto \exp\{\xi_0 \cdot T(\Theta)\}$; that is, $p(\Theta)$ is an exponential family distribution for $\Theta$ with sufficient statistic $T(\Theta)$ and natural parameter $\xi_0$. We suppose further that $\mathcal{A}$ always returns a distribution in the same exponential family; in particular, we suppose that there exists some parameter $\xi_b$ such that

$$q_b(\Theta) \propto \exp\{\xi_b \cdot T(\Theta)\} \quad \text{for} \quad q_b(\Theta) = \mathcal{A}(C_b, p(\Theta)). \tag{5}$$

When we make these two assumptions, the update in Eq. (4) becomes

$$p(\Theta \mid C_1, \ldots, C_B) \approx q(\Theta) \propto \exp\left\{ \left[\xi_0 + \sum_{b=1}^{B}(\xi_b - \xi_0)\right] \cdot T(\Theta) \right\}, \tag{6}$$

where the normalizing constant is readily obtained from the exponential family form. In what follows we use the shorthand $\xi \leftarrow \mathcal{A}(C, \xi_0)$ to denote that $\mathcal{A}$ takes as input a minibatch $C$ and a prior with exponential family parameter $\xi_0$ and that it returns a distribution in the same exponential family with parameter $\xi$.

So, to approximate $p(\Theta \mid C_1, \ldots, C_B)$, we first calculate $\xi_b$ via the approximation primitive $\mathcal{A}$ for each minibatch $C_b$; note that these calculations may be performed in parallel. Then we sum together the quantities $\xi_b - \xi_0$ across $b$, along with the initial $\xi_0$ from the prior, to find the final exponential family parameter to the full posterior approximation $q$. We previously saw that the general Bayes sequential update can be made streaming by iterating with the old posterior as the new prior (Eq. (2)). Similarly, here we see that the full posterior approximation $q$ is in the same exponential family as the prior, so one may iterate these parallel computations to arrive at a parallelized algorithm for streaming posterior computation.

We emphasize that while these updates are reminiscent of prior-posterior conjugacy, it is actually the approximate posteriors and single, original prior that we assume belong to the same exponential family. It is not necessary to assume any conjugacy in the generative model itself nor that any true intermediate or final posterior take any particular limited form.

**Asynchronous Bayesian updating.** Performing $B$ computations in parallel can in theory speed up algorithm running time by a factor of $B$, but in practice it is often the case that a single computation thread takes longer than the rest. Waiting for this thread to finish diminishes potential gains from distributing the computations. This problem can be ameliorated by making computations *asynchronous*. In this case, processors known as *workers* each solve a subproblem. When a worker finishes, it reports its solution to a single *master* processor. If the master gives the worker a new subproblem without waiting for the other workers to finish, it can decrease downtime in the system.

Our asynchronous algorithm is in the spirit of Hogwild! [1]. To present the algorithm we first describe an asynchronous computation that we will not use in practice, but which will serve as a conceptual stepping stone. Note in particular that the following scheme makes the computations in Eq. (6) asynchronous. Have each worker continuously iterate between three steps: (1) collect a new minibatch $C$, (2) compute the local approximate posterior $\xi \leftarrow \mathcal{A}(C, \xi_0)$, and (3) return $\Delta\xi := \xi - \xi_0$ to the master. The master, in turn, starts by assigning the posterior to equal the prior: $\xi^{(\text{post})} \leftarrow \xi_0$. Each time the master receives a quantity $\Delta\xi$ from any worker, it updates the posterior synchronously: $\xi^{(\text{post})} \leftarrow \xi^{(\text{post})} + \Delta\xi$. If $\mathcal{A}$ returns the exponential family parameter of the true posterior (rather than an approximation), then the posterior at the master is exact by Eq. (4).

A preferred asynchronous computation works as follows. The master initializes its posterior estimate to the prior: $\xi^{(\text{post})} \leftarrow \xi_0$. Each worker continuously iterates between four steps: (1) collect a new minibatch $C$, (2) copy the master posterior value locally $\xi^{(\text{local})} \leftarrow \xi^{(\text{post})}$, (3) compute the local approximate posterior $\xi \leftarrow \mathcal{A}(C, \xi^{(\text{local})})$, and (4) return $\Delta\xi := \xi - \xi^{(\text{local})}$ to the master. Each time the master receives a quantity $\Delta\xi$ from any worker, it updates the posterior synchronously: $\xi^{(\text{post})} \leftarrow \xi^{(\text{post})} + \Delta\xi$.

The key difference between the first and second frameworks proposed above is that, in the second, the latest posterior is used as a prior. This latter framework is more in line with the streaming update of Eq. (2) but introduces a new layer of approximation. Since $\xi^{(\text{post})}$ might change at the master

while the worker is computing $\Delta\xi$, it is no longer the case that the posterior at the master is exact when $\mathcal{A}$ returns the exponential family parameter of the true posterior. Nonetheless we find that the latter framework performs better in practice, so we focus on it exclusively in what follows.

We refer to our overall framework as *SDA-Bayes*, which stands for (S)treaming, (D)istributed, (A)synchronous Bayes. The framework is intended to be general enough to allow a variety of local approximations $\mathcal{A}$. Indeed, SDA-Bayes works out of the box once an implementation of $\mathcal{A}$—and a prior on the global parameter(s) $\Theta$—is provided. In the current paper our preferred local approximation will be VB.

# 3  Case study: latent Dirichlet allocation

In what follows, we consider examples of the choices for the $\Theta$ prior and primitive $\mathcal{A}$ in the context of *latent Dirichlet allocation* (LDA) [13]. LDA models the content of $D$ documents in a corpus. Themes potentially shared by multiple documents are described by *topics*. The unsupervised learning problem is to learn the topics as well as discover which topics occur in which documents.

More formally, each topic (of $K$ total topics) is a distribution over the $V$ words in the vocabulary: $\beta_k = (\beta_{kv})_{v=1}^V$. Each document is an admixture of topics. The words in document $d$ are assumed to be exchangeable. Each word $w_{dn}$ belongs to a latent topic $z_{dn}$ chosen according to a document-specific distribution of topics $\theta_d = (\theta_{dk})_{k=1}^K$. The full generative model, with Dirichlet priors for $\beta_k$ and $\theta_d$ conditioned on respective parameters $\eta_k$ and $\alpha$, appears in [13].

To see that this model fits our specification in Sec. 2, consider the set of global parameters $\Theta = \beta$. Each document $w_d = (w_{dn})_{n=1}^{N_d}$ is distributed iid conditioned on the global topics. The full collection of data is a corpus $C = w = (w_d)_{d=1}^D$ of documents. The posterior for LDA, $p(\beta, \theta, z \mid C, \eta, \alpha)$, is equal to the following expression up to proportionality:

$$\propto \left[ \prod_{k=1}^K \text{Dirichlet}(\beta_k \mid \eta_k) \right] \cdot \left[ \prod_{d=1}^D \text{Dirichlet}(\theta_d \mid \alpha) \right] \cdot \left[ \prod_{d=1}^D \prod_{n=1}^{N_d} \theta_{dz_{dn}} \beta_{z_{dn}, w_{dn}} \right]. \qquad (7)$$

The posterior for just the global parameters $p(\beta \mid C, \eta, \alpha)$ can be obtained from $p(\beta, \theta, z \mid C, \eta, \alpha)$ by integrating out the local, document-specific parameters $\theta, z$. As is common in complex models, the normalizing constant for Eq. (7) is intractable to compute, so the posterior must be approximated.

## 3.1  Posterior-approximation algorithms

To apply SDA-Bayes to LDA, we use the prior specified by the generative model. It remains to choose a posterior-approximation algorithm $\mathcal{A}$. We consider two possibilities here: variational Bayes (VB) and expectation propagation (EP). Both primitives take Dirichlet distributions as priors for $\beta$ and both return Dirichlet distributions for the approximate posterior of the topic parameters $\beta$; thus the prior and approximate posterior are in the same exponential family. Hence both VB and EP can be utilized as a choice for $\mathcal{A}$ in the SDA-Bayes framework.

**Mean-field variational Bayes.** We use the shorthand $p_D$ for Eq. (7), the posterior given $D$ documents. We assume the approximating distribution, written $q_D$ for shorthand, takes the form

$$q_D(\beta, \theta, z \mid \lambda, \gamma, \phi) = \left[ \prod_{k=1}^K q_D(\beta_k \mid \lambda_k) \right] \cdot \left[ \prod_{d=1}^D q_D(\theta_d \mid \gamma_d) \right] \cdot \left[ \prod_{d=1}^D \prod_{n=1}^{N_d} q_D(z_{dn} \mid \phi_{dw_{dn}}) \right] \qquad (8)$$

for parameters $(\lambda_{kv}), (\gamma_{dk}), (\phi_{dvk})$ with $k \in \{1, \dots, K\}, v \in \{1, \dots, V\}, d \in \{1, \dots, D\}$. Moreover, we set $q_D(\beta_k \mid \lambda_k) = \text{Dirichlet}_V(\beta_k \mid \lambda_k)$, $q_D(\theta_d \mid \gamma_d) = \text{Dirichlet}_K(\theta_d \mid \gamma_d)$, and $q_D(z_{dn} \mid \phi_{dw_{dn}}) = \text{Categorical}_K(z_{dn} \mid \phi_{dw_{dn}})$. The subscripts on Dirichlet and Categorical indicate the dimensions of the distributions (and of the parameters).

The problem of VB is to find the best approximating $q_D$, defined as the collection of variational parameters $\lambda, \gamma, \phi$ that minimize the KL divergence from the true posterior: $\text{KL}(q_D \| p_D)$. Even finding the minimizing parameters is a difficult optimization problem. Typically the solution is approximated by coordinate descent in each parameter [6, 13] as in Alg. 1. The derivation of VB for LDA can be found in [4, 13] and Sup. Mat. A.1.

**Algorithm 1:** VB for LDA

**Input**: Data $(n_d)_{d=1}^D$; hyperparameters $\eta, \alpha$
**Output**: $\lambda$
Initialize $\lambda$
**while** $(\lambda, \gamma, \phi)$ *not converged* **do**
  **for** $d = 1, \ldots, D$ **do**
    $(\gamma_d, \phi_d) \leftarrow \texttt{LocalVB}(d, \lambda)$
  $\forall(k, v), \lambda_{kv} \leftarrow \eta_{kv} + \sum_{d=1}^D \phi_{dvk} n_{dv}$

---

**Subroutine** $\texttt{LocalVB}(d, \lambda)$

  **Output**: $(\gamma_d, \phi_d)$
  Initialize $\gamma_d$
  **while** $(\gamma_d, \phi_d)$ *not converged* **do**
    $\forall(k, v)$, set $\phi_{dvk} \propto$
    $\exp\left(\mathbb{E}_q[\log \theta_{dk}] + \mathbb{E}_q[\log \beta_{kv}]\right)$
    (normalized across $k$)
    $\forall k, \gamma_{dk} \leftarrow \alpha_k + \sum_{v=1}^V \phi_{dvk} n_{dv}$

---

**Algorithm 2:** SVI for LDA

**Input**: Hyperparameters $\eta, \alpha, D, (\rho_t)_{t=1}^T$
**Output**: $\lambda$
Initialize $\lambda$
**for** $t = 1, \ldots, T$ **do**
  Collect new data minibatch $C$
  **foreach** *document indexed $d$ in $C$* **do**
    $(\gamma_d, \phi_d) \leftarrow \texttt{LocalVB}(d, \lambda)$
  $\forall(k, v), \tilde{\lambda}_{kv} \leftarrow \eta_{kv} + \frac{D}{|C|} \sum_{d \text{ in } C} \phi_{dvk} n_{dv}$
  $\forall(k, v), \lambda_{kv} \leftarrow (1 - \rho_t) \lambda_{kv} + \rho_t \tilde{\lambda}_{kv}$

---

**Algorithm 3:** SSU for LDA

**Input**: Hyperparameters $\eta, \alpha$
**Output**: A sequence $\lambda^{(1)}, \lambda^{(2)}, \ldots$
Initialize $\forall(k, v), \lambda_{kv}^{(0)} \leftarrow \eta_{kv}$
**for** $b = 1, 2, \ldots$ **do**
  Collect new data minibatch $C$
  **foreach** *document indexed $d$ in $C$* **do**
    $(\gamma_d, \phi_d) \leftarrow \texttt{LocalVB}(d, \lambda)$
  $\forall(k, v), \lambda_{kv}^{(b)} \leftarrow \lambda_{kv}^{(b-1)} + \sum_{d \text{ in } C} \phi_{dvk} n_{dv}$

---

Figure 1: Algorithms for calculating $\lambda$, the parameters for the topic posteriors in LDA. VB iterates multiple times through the data, SVI makes a single pass, and SSU is streaming. Here, $n_{dv}$ represents the number of words $v$ in document $d$.

**Expectation propagation.** An EP [7] algorithm for approximating the LDA posterior appears in Alg. 6 of Sup. Mat. B. Alg. 6 differs from [14], which does not provide an approximate posterior for the topic parameters, and is instead our own derivation. Our version of EP, like VB, learns factorized Dirichlet distributions over topics.

### 3.2 Other single-pass algorithms for approximate LDA posteriors

The algorithms in Sec. 3.1 pass through the data multiple times and require storing the data set in memory—but are useful as primitives for SDA-Bayes in the context of the processing of minibatches of data. Next, we consider two algorithms that can pass through a data set just one time (*single pass*) and to which we compare in the evaluations (Sec. 4).

**Stochastic variational inference.** VB uses coordinate descent to find a value of $q_D$, Eq. (8), that locally minimizes the KL divergence, $\text{KL}(q_D \| p_D)$. *Stochastic variational inference* (SVI) [3, 4] is exactly the application of a particular version of stochastic gradient descent to the same optimization problem. While stochastic gradient descent can often be viewed as a streaming algorithm, the optimization problem itself here depends on $D$ via $p_D$, the posterior on $D$ data points. We see that, as a result, $D$ must be specified in advance, appears in each step of SVI (see Alg. 2), and is independent of the number of data points actually processed by the algorithm. Nonetheless, while one may choose to visit $D' \neq D$ data points or revisit data points when using SVI to estimate $p_D$ [3, 4], SVI can be made single-pass by visiting each of $D$ data points exactly once and then has constant memory requirements. We also note that two new parameters, $\tau_0 > 0$ and $\kappa \in (0.5, 1]$, appear in SVI, beyond those in VB, to determine a learning rate $\rho_t$ as a function of iteration $t$: $\rho_t := (\tau_0 + t)^{-\kappa}$.

**Sufficient statistics.** On each round of VB (Alg. 1), we update the local parameters for all documents and then compute $\lambda_{kv} \leftarrow \eta_{kv} + \sum_{d=1}^D \phi_{dvk} n_{dv}$. An alternative single-pass (and indeed streaming) option would be to update the local parameters for each minibatch of documents as they arrive and then add the corresponding terms $\phi_{dvk} n_{dv}$ to the current estimate of $\lambda$ for each document $d$ in the minibatch. This essential idea has been proposed previously for models other than LDA by [11, 12] and forms the basis of what we call the *sufficient statistics update algorithm* (SSU): Alg. 3. This algorithm is equivalent to SDA-Bayes with $\mathcal{A}$ chosen to be a single iteration over the global variable $\lambda$ of VB (i.e., updating $\lambda$ exactly once instead of iterating until convergence).

|  | Wikipedia | | | | Nature | | | |
|---|---|---|---|---|---|---|---|---|
|  | 32-SDA | 1-SDA | SVI | SSU | 32-SDA | 1-SDA | SVI | SSU |
| Log pred prob | **−7.31** | −7.43 | −7.32 | −7.91 | −7.11 | −7.19 | **−7.08** | −7.82 |
| Time (hours) | **2.09** | 43.93 | 7.87 | 8.28 | **0.55** | 10.02 | 1.22 | 1.27 |

Table 1: A comparison of (1) log predictive probability of held-out data and (2) running time of four algorithms: SDA-Bayes with 32 threads, SDA-Bayes with 1 thread, SVI, and SSU.

## 4 Evaluation

We follow [4] (and further [15, 16]) in evaluating our algorithms by computing (approximate) predictive probability. Under this metric, a higher score is better, as a better model will assign a higher probability to the held-out words.

We calculate predictive probability by first setting aside held-out testing documents $C^{(\text{test})}$ from the full corpus and then further setting aside a subset of held-out testing words $W_{d,\text{test}}$ in each testing document $d$. The remaining (training) documents $C^{(\text{train})}$ are used to estimate the global parameter posterior $q(\beta)$, and the remaining (training) words $W_{d,\text{train}}$ within the $d$th testing document are used to estimate the document-specific parameter posterior $q(\theta_d)$.[1] To calculate predictive probability, an approximation is necessary since we do not know the predictive distribution—just as we seek to learn the posterior distribution. Specifically, we calculate the normalized predictive distribution and report "log predictive probability" as

$$\frac{\sum_{d \in C^{(\text{test})}} \log p(W_{d,\text{test}} \mid C^{(\text{train})}, W_{d,\text{train}})}{\sum_{d \in C^{(\text{test})}} |W_{d,\text{test}}|} = \frac{\sum_{d \in C^{(\text{test})}} \sum_{w_{\text{test}} \in W_{d,\text{test}}} \log p(w_{\text{test}} \mid C^{(\text{train})}, W_{d,\text{train}})}{\sum_{d \in C^{(\text{test})}} |W_{d,\text{test}}|},$$

where we use the approximation

$$p(w_{\text{test}} \mid C^{(\text{train})}, W_{d,\text{train}}) = \int_\beta \int_{\theta_d} \left( \sum_{k=1}^K \theta_{dk} \beta_{k w_{\text{test}}} \right) p(\theta_d \mid W_{d,\text{train}}, \beta) \, p(\beta \mid C^{(\text{train})}) \, d\theta_d \, d\beta$$

$$\approx \int_\beta \int_{\theta_d} \left( \sum_{k=1}^K \theta_{dk} \beta_{k w_{\text{test}}} \right) q(\theta_d) \, q(\beta) \, d\theta_d \, d\beta = \sum_{k=1}^K \mathbb{E}_q[\theta_{dk}] \, \mathbb{E}_q[\beta_{k w_{\text{test}}}].$$

To facilitate comparison with SVI, we use the Wikipedia and Nature corpora of [3, 5] in our experiments. These two corpora represent a range of sizes (3,611,558 training documents for Wikipedia and 351,525 for Nature) as well as different types of topics. We expect words in Wikipedia to represent an extremely broad range of topics whereas we expect words in Nature to focus more on the sciences. We further use the vocabularies of [3, 5] and SVI code available online at [17]. We hold out 10,000 Wikipedia documents and 1,024 Nature documents (not included in the counts above) for testing. In the results presented in the main text, we follow [3, 4] in fitting an LDA model with $K = 100$ topics and hyperparameters chosen as: $\forall k, \alpha_k = 1/K, \forall(k,v), \eta_{kv} = 0.01$. For both Wikipedia and Nature, we set the parameters in SVI according to the optimal values of the parameters described in Table 1 of [3] (number of documents $D$ correctly set in advance, step size parameters $\kappa = 0.5$ and $\tau_0 = 64$).

Figs. 3(a) and 3(d) demonstrate that both SVI and SDA are sensitive to minibatch size when $\eta_{kv} = 0.01$, with generally superior performance at larger batch sizes. Interestingly, both SVI and SDA performance improve and are steady across batch size when $\eta_{kv} = 1$ (Figs. 3(a) and 3(d)). Nonetheless, we use $\eta_{kv} = 0.01$ in what follows in the interest of consistency with [3, 4]. Moreover, in the remaining experiments, we use a large minibatch size of $2^{15} = 32,768$. This size is the largest before SVI performance degrades in the Nature data set (Fig. 3(d)).

Performance and timing results are shown in Table 1. One would expect that with additional streaming capabilities, SDA-Bayes should show a performance loss relative to SVI. We see from Table 1

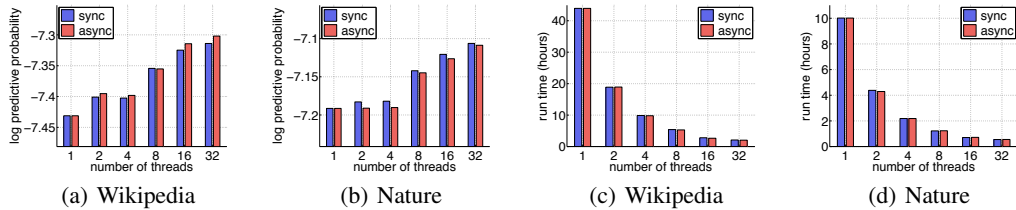

|     (a) Wikipedia     |     (b) Nature     |     (c) Wikipedia     |     (d) Nature     |

Figure 2: SDA-Bayes log predictive probability (*two left plots*) and run time (*two right plots*) as a function of number of threads.

that such loss is small in the single-thread case, while SSU performs much worse. SVI is faster than single-thread SDA-Bayes in this single-pass setting.

**Full SDA-Bayes improves run time with no performance cost.** We handicap SDA-Bayes in the above comparisons by utilizing just a single thread. In Table 1, we also report performance of SDA-Bayes with 32 threads and the same minibatch size. In the synchronous case, we consider minibatch size to equal the total number of data points processed per round; therefore, the minibatch size equals the number of data points sent to each thread per round times the total number of threads. In the asynchronous case, we analogously report minibatch size as this product.

Fig. 2 shows the performance of SDA-Bayes when we run with $\{1, 2, 4, 8, 16, 32\}$ threads while keeping the minibatch size constant. The goal in such a distributed context is to improve run time while not hurting performance. Indeed, we see dramatic run time improvement as the number of threads grows and in fact some slight performance improvement as well. We tried both a parallel version and a full distributed, asynchronous version of the algorithm; Fig. 2 indicates that the speedup and performance improvements we see here come from parallelizing—which is theoretically justified by Eq. (3) when $\mathcal{A}$ is Bayes rule. Our experiments indicate that our Hogwild!-style asynchrony does not hurt performance. In our experiments, the processing time at each thread seems to be approximately equal across threads and dominate any communication time at the master, so synchronous and asynchronous performance and running time are essentially identical. In general, a practitioner might prefer asynchrony since it is more robust to node failures.

**SVI is sensitive to the choice of total data size $D$.** The evaluations above are for a single posterior over $D$ data points. Of greater concern to us in this work is the evaluation of algorithms in the streaming setting. We have seen that SVI is designed to find the posterior for a particular, pre-chosen number of data points $D$. In practice, when we run SVI on the full data set but change the input value of $D$ in the algorithm, we can see degradations in performance. In particular, we try values of $D$ equal to $\{0.01, 0.1, 1, 10, 100\}$ times the true $D$ in Fig. 3(b) for the Wikipedia data set and in Fig. 3(e) for the Nature data set.

A practitioner in the streaming setting will typically not know $D$ in advance, or multiple values of $D$ may be of interest. Figs. 3(b) and 3(e) illustrate that an estimate may not be sufficient. Even in the case where $D$ is known in advance, it is reasonable to imagine a new influx of further data. One might need to run SVI again from the start (and, in so doing, revisit the first data set) to obtain the desired performance.

**SVI is sensitive to learning step size.** [3, 5] use cross-validation to tune step-size parameters $(\tau_0, \kappa)$ in the stochastic gradient descent component of the SVI algorithm. This cross-validation requires multiple runs over the data and thus is not suited to the streaming setting. Figs. 3(c) and 3(f) demonstrate that the parameter choice does indeed affect algorithm performance. In these figures, we keep $D$ at the true training data size.

[3] have observed that the optimal $(\tau_0, \kappa)$ may interact with minibatch size, and we further observe that the optimal values may vary with $D$ as well. We also note that recent work has suggested a way to update $(\tau_0, \kappa)$ adaptively during an SVI run [18].

**EP is not suited to LDA.** Earlier attempts to apply EP to the LDA model in the non-streaming setting have had mixed success, with [19] in particular finding that EP performance can be poor for LDA and, moreover, that EP requires "unrealistic intermediate storage requirements." We found

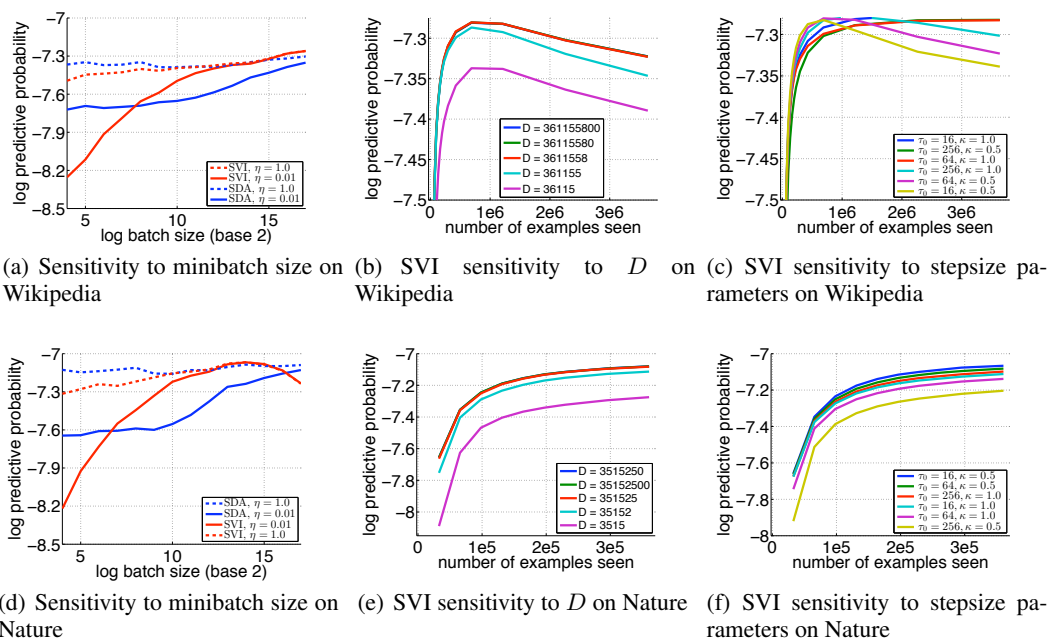

(a) Sensitivity to minibatch size on Wikipedia
(b) SVI sensitivity to $D$ on Wikipedia
(c) SVI sensitivity to stepsize parameters on Wikipedia
(d) Sensitivity to minibatch size on Nature
(e) SVI sensitivity to $D$ on Nature
(f) SVI sensitivity to stepsize parameters on Nature

Figure 3: Sensitivity of SVI and SDA-Bayes to some respective parameters. Legends have the same top-to-bottom order as the rightmost curve points.

this to also be true in the streaming setting. We were not able to obtain competitive results with EP; based on an 8-thread implementation of SDA-Bayes with an EP primitive[2], after over 91 hours on Wikipedia (and $6.7 \times 10^4$ data points), log predictive probability had stabilized at around $-7.95$ and, after over 97 hours on Nature (and $9.7 \times 10^4$ data points), log predictive probability had stabilized at around $-8.02$. Although SDA-Bayes with the EP primitive is not effective for LDA, it remains to be seen whether this combination may be useful in other domains where EP is known to be effective.

## 5 Discussion

We have introduced SDA-Bayes, a framework for streaming, distributed, asynchronous computation of an approximate Bayesian posterior. Our framework makes streaming updates to the estimated posterior according to a user-specified approximation primitive. We have demonstrated the usefulness of our framework, with variational Bayes as the primitive, by fitting the latent Dirichlet allocation topic model to the Wikipedia and Nature corpora. We have demonstrated the advantages of our algorithm over stochastic variational inference and the sufficient statistics update algorithm, particularly with respect to the key issue of obtaining approximations to posterior probabilities based on the number of documents seen thus far, not posterior probabilities for a fixed number of documents.

#### Acknowledgments

We thank M. Hoffman, C. Wang, and J. Paisley for discussions, code, and data and our reviewers for helpful comments. TB is supported by the Berkeley Fellowship, NB by a Hertz Foundation Fellowship, and ACW by the Chancellor's Fellowship at UC Berkeley. This research is supported in part by NSF award CCF-1139158, DARPA Award FA8750-12-2-0331, AMPLab sponsor donations, and the ONR under grant number N00014-11-1-0688.

## Footnotes

[1] In all cases, we estimate $q(\theta_d)$ for evaluative purposes using VB since direct EP estimation takes prohibitively long.

[2]We chose 8 threads since any fewer was too slow to get results and anything larger created too high of a memory demand on our system.

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
