[Supplementary Material]

SUPPLEMENTARY MATERIAL TO
# Streaming Variational Bayes

**Tamara Broderick,   Nicholas Boyd,   Andre Wibisono,   Ashia C. Wilson**
University of California, Berkeley
{tab@stat, nickboyd@eecs, wibisono@eecs, ashia@stat}.berkeley.edu

**Michael I. Jordan**
University of California, Berkeley
jordan@cs.berkeley.edu

## A  Variational Bayes

### A.1  Batch VB

As described in the main text, the idea of VB is to find the distribution $q_D$ that best approximates the true posterior, $p_D$. More specifically, the optimization problem of VB is defined as finding a $q_D$ to minimize the KL divergence between the approximating distribution and the posterior:

$$\text{KL}\left(q_D \parallel p_D\right) := \mathbb{E}_{q_D}\left[\log\left(q_D/p_D\right)\right]$$

Typically $q_D$ takes a particular, constrained form, and finding the optimal $q_D$ amounts to finding the optimal parameters for $q_D$. Moreover, the optimal parameters usually cannot be expressed in closed form, so often a coordinate descent algorithm is used.

For the LDA model, we have $q_D$ in the form of Eq. (8) and $p_D$ defined by Eq. (7). We wish to find the following variational parameters (i.e., parameters to $q_D$): $\lambda$ (describing each topic), $\gamma$ (describing the topic proportions in each document), and $\phi$ (describing the assignment of each word in each document to a topic).

#### A.1.1  Evidence lower bound

Finding $q_D$ to minimize the KL divergence between $q_D$ and $p_D$ is equivalent to finding $q_D$ to maximize the *evidence lower bound* (ELBO),

$$
\begin{aligned}
\text{ELBO} &:= \mathbb{E}_{q_D}\left[\log p(\Theta, x_{1:D})\right] - \mathbb{E}_{q_D}\left[\log q_D\right] \\
&= \mathbb{E}_{q_D}\left[\log p_D\right] + p(x_{1:D}) - \mathbb{E}_{q_D}\left[\log q_D\right] \\
&= -\text{KL}\left(q_D \parallel p_D\right) + p(x_{1:D}),
\end{aligned}
$$

since $p(x_{1:D})$ is constant in $q_D$. The VB optimization problem is often phrased in terms of the ELBO instead of the KL divergence.

The ELBO for LDA can be written as follows, where the model parameters are $\beta, \theta, z$ and the data is $w$; $\eta$ and $\alpha$ are fixed hyperparameters.

$$
\begin{aligned}
\text{ELBO}(\lambda, \gamma, \phi) &= \mathbb{E}_q\left[\log p(\beta, \theta, z, w \mid \eta, \alpha)\right] - \mathbb{E}_q\left[\log q(\beta, \theta, z \mid \lambda, \gamma, \phi)\right] \\
&= \sum_{k=1}^{K} \mathbb{E}_q\left[\log \text{Dirichlet}(\beta_k \mid \eta_k)\right] + \sum_{d=1}^{D} \mathbb{E}_q\left[\log \text{Dirichlet}(\theta_d \mid \alpha)\right] \\
&\quad + \sum_{d=1}^{D}\sum_{n=1}^{N_d} \mathbb{E}_q\left[\log \text{Multinomial}(z_{dn} \mid \theta_d)\right] + \sum_{d=1}^{D}\sum_{n=1}^{N_d} \mathbb{E}_q\left[\log \text{Multinomial}(w_{dn} \mid \beta_{z_{dn}})\right]
\end{aligned}
$$

$$-\sum_{k=1}^{K} \mathbb{E}_q\left[\log \text{Dirichlet}(\beta_k \mid \lambda_k)\right] - \sum_{d=1}^{D} \mathbb{E}_q\left[\log \text{Dirichlet}(\theta_d \mid \gamma_d)\right]$$

$$-\sum_{d=1}^{D}\sum_{n=1}^{N_d} \mathbb{E}_q\left[\log \text{Multinomial}(z_{dn} \mid \phi_{dw_{dn}})\right].$$

The expectations in $q$ in the previous equation can be evaluated as follows. The equations below make use of the *digamma function* $\psi$ and *trigamma function* $\psi_1$. Here,

$$\psi(x) = \frac{d}{dx} \log \Gamma(x) = \left[\frac{d}{dx}\Gamma(x)\right] / \Gamma(x)$$

$$\psi_1(x) = \frac{d^2}{dx^2} \log \Gamma(x) = \frac{d}{dx}\psi(x).$$

Then,

$\mathbb{E}_q\left[\log \text{Dirichlet}(\beta_k \mid \eta_k)\right]$

$$= \log \Gamma\left(\sum_{v=1}^{V} \eta_{kv}\right) - \sum_{v=1}^{V} \log \Gamma(\eta_{kv}) + \sum_{v=1}^{V}(\eta_{kv} - 1)\, \mathbb{E}_q[\log \beta_{kv}]$$

$$= \log \Gamma\left(\sum_{v=1}^{V} \eta_{kv}\right) - \sum_{v=1}^{V} \log \Gamma(\eta_{kv}) + \sum_{v=1}^{V}(\eta_{kv} - 1)\left(\psi(\lambda_{kv}) - \psi\left(\sum_{u=1}^{V} \lambda_{ku}\right)\right)$$

$\mathbb{E}_q\left[\log \text{Dirichlet}(\theta_d \mid \alpha)\right]$

$$= \log \Gamma\left(\sum_{k=1}^{K} \alpha_k\right) - \sum_{k=1}^{K} \log \Gamma(\alpha_k) + \sum_{k=1}^{K}(\alpha_k - 1)\, \mathbb{E}_q[\log \theta_{dk}]$$

$$= \log \Gamma\left(\sum_{k=1}^{K} \alpha_k\right) - \sum_{k=1}^{K} \log \Gamma(\alpha_k) + \sum_{k=1}^{K}(\alpha_k - 1)\left(\psi(\gamma_{dk}) - \psi\left(\sum_{j=1}^{K} \gamma_{dj}\right)\right)$$

$\mathbb{E}_q\left[\log \text{Multinomial}(z_{dn} \mid \theta_d)\right]$

$$= \sum_{k=1}^{K} \phi_{dw_{dn}k}\mathbb{E}_q[\log \theta_{dk}]$$

$$= \sum_{k=1}^{K} \phi_{dw_{dn}k}\left(\psi(\gamma_{dk}) - \psi\left(\sum_{j=1}^{K} \gamma_{dj}\right)\right)$$

$\mathbb{E}_q\left[\log \text{Multinomial}(w_{dn} \mid \beta_{z_{dn}})\right]$

$$= \sum_{v=1}^{V} \mathbb{1}\{w_{dn} = v\}\, \mathbb{E}_q[\log \beta_{z_{dn},v}]$$

$$= \sum_{v=1}^{V} \mathbb{1}\{w_{dn} = v\} \sum_{k=1}^{K} \phi_{dw_{dn}k}\mathbb{E}_q[\log \beta_{kv}]$$

$$= \sum_{v=1}^{V}\sum_{k=1}^{K} \mathbb{1}\{w_{dn} = v\}\, \phi_{dw_{dn}k}\left(\psi(\lambda_{kv}) - \psi\left(\sum_{u=1}^{V} \lambda_{ku}\right)\right)$$

$\mathbb{E}_q\left[\log \text{Dirichlet}(\beta_k \mid \lambda_k)\right]$

$$= \log \Gamma\left(\sum_{v=1}^{V} \lambda_{kv}\right) - \sum_{v=1}^{V} \log \Gamma(\lambda_{kv}) + \sum_{v=1}^{V}(\lambda_{kv} - 1)\, \mathbb{E}_q[\log \beta_{kv}]$$

$$= \log \Gamma\left(\sum_{v=1}^{V} \lambda_{kv}\right) - \sum_{v=1}^{V} \log \Gamma(\lambda_{kv}) + \sum_{v=1}^{V}(\lambda_{kv} - 1)\left(\psi(\lambda_{kv}) - \psi\left(\sum_{u=1}^{V} \lambda_{ku}\right)\right)$$

$\mathbb{E}_q\left[\log \text{Dirichlet}(\theta_d \mid \gamma_d)\right]$

$$= \log \Gamma \left( \sum_{k=1}^{K} \gamma_{dk} \right) - \sum_{k=1}^{K} \log \Gamma(\gamma_{dk}) + \sum_{k=1}^{K} (\gamma_{dk} - 1) \, \mathbb{E}_q[\log \theta_{dk}]$$

$$= \log \Gamma \left( \sum_{k=1}^{K} \gamma_{dk} \right) - \sum_{k=1}^{K} \log \Gamma(\gamma_{dk}) + \sum_{k=1}^{K} (\gamma_{dk} - 1) \left( \psi(\gamma_{dk}) - \psi\left( \sum_{j=1}^{K} \gamma_{dj} \right) \right)$$

$$\mathbb{E}_q \left[ \log \mathrm{Multinomial}(z_{dn} \mid \phi_{dn}) \right]$$

$$= \sum_{k=1}^{K} \phi_{dw_{dn}k} \log \phi_{dw_{dn}k}.$$

### A.1.2 Coordinate ascent

We maximize the ELBO via coordinate ascent in each dimension of the variational parameters: $\lambda$, $\gamma$, and $\phi$.

**Variational parameter $\lambda$.** Choose a topic index $k$. Fix $\gamma$, $\phi$, and each $\lambda_j$ for $j \neq k$. Then we can write the ELBO's functional dependence on $\lambda_k$ as follows, where "const" is a constant in $\lambda_k$.

$$
\begin{aligned}
\mathrm{ELBO}(\lambda_k) =& \sum_{v=1}^{V} (\eta_{kv} - 1) \left( \psi(\lambda_{kv}) - \psi\left( \sum_{u=1}^{V} \lambda_{ku} \right) \right) \\
&+ \sum_{d=1}^{D} \sum_{n=1}^{N_d} \sum_{v=1}^{V} \mathbb{1}\{w_{dn} = v\} \, \phi_{dw_{dn}k} \left( \psi(\lambda_{kv}) - \psi\left( \sum_{u=1}^{V} \lambda_{ku} \right) \right) \\
&- \log \Gamma \left( \sum_{v=1}^{V} \lambda_{kv} \right) + \sum_{v=1}^{V} \log \Gamma(\lambda_{kv}) \\
&- \sum_{v=1}^{V} (\lambda_{kv} - 1) \left( \psi(\lambda_{kv}) - \psi\left( \sum_{u=1}^{V} \lambda_{ku} \right) \right) + \mathrm{const} \\
=& \sum_{v=1}^{V} \left( \eta_{kv} - \lambda_{kv} + \sum_{d=1}^{D} \sum_{n=1}^{N_d} \mathbb{1}\{w_{dn} = v\} \, \phi_{dw_{dn}k} \right) \left( \psi(\lambda_{kv}) - \psi\left( \sum_{u=1}^{V} \lambda_{ku} \right) \right) \\
&- \log \Gamma \left( \sum_{v=1}^{V} \lambda_{kv} \right) + \sum_{v=1}^{V} \log \Gamma(\lambda_{kv}) + \mathrm{const}
\end{aligned}
$$

The partial derivative of $\mathrm{ELBO}(\lambda_k)$ with respect to one of the dimensions of $\lambda_k$, say $\lambda_{kv}$, is

$$
\begin{aligned}
&\frac{\partial}{\partial \lambda_{kv}} \mathrm{ELBO}(\lambda_k) \\
=& - \left( \psi(\lambda_{kv}) - \psi\left( \sum_{u=1}^{V} \lambda_{ku} \right) \right) \\
&+ \left( \eta_{kv} - \lambda_{kv} + \sum_{d=1}^{D} \sum_{n=1}^{N_d} \mathbb{1}\{w_{dn} = v\} \, \phi_{dw_{dn}k} \right) \left( \psi_1(\lambda_{kv}) - \psi_1\left( \sum_{u=1}^{V} \lambda_{ku} \right) \right) \\
&- \sum_{t:t \neq v} \left( \eta_{kt} - \lambda_{kt} + \sum_{d=1}^{D} \sum_{n=1}^{N_d} \mathbb{1}\{w_{dn} = t\} \, \phi_{dw_{dn}k} \right) \psi_1 \left( \sum_{u=1}^{V} \lambda_{ku} \right) - \psi\left( \sum_{u=1}^{V} \lambda_{ku} \right) + \psi(\lambda_{kv}) \\
=& \; \psi_1(\lambda_{kv}) \left( \eta_{kv} - \lambda_{kv} + \sum_{d=1}^{D} \sum_{n=1}^{N_d} \mathbb{1}\{w_{dn} = v\} \, \phi_{dw_{dn}k} \right) \\
&- \psi\left( \sum_{u=1}^{V} \lambda_{ku} \right) \sum_{u=1}^{V} \left( \eta_{ku} - \lambda_{ku} + \sum_{d=1}^{D} \sum_{n=1}^{N_d} \mathbb{1}\{w_{dn} = u\} \, \phi_{dw_{dn}k} \right).
\end{aligned}
$$

From the last line of the previous equation, we see that one can set the gradient of $\mathrm{ELBO}(\lambda_k)$ to zero by setting

$$\lambda_{kv} \leftarrow \eta_{kv} + \sum_{d=1}^{D} \sum_{n=1}^{N_d} \mathbb{1}\{w_{dn} = v\}\, \phi_{dw_{dn}k} \quad \text{for } v = 1, \ldots, V.$$

Equivalently, if $n_{dv}$ is the number of occurrences (tokens) of word type $v$ in document $d$, then the update may be written

$$\lambda_{kv} \leftarrow \eta_{kv} + \sum_{d=1}^{D} n_{dv}\, \phi_{dvk} \quad \text{for } v = 1, \ldots, V.$$

**Variational parameter $\gamma$.** Now choose a document $d$. Fix $\lambda$, $\phi$, and $\gamma_c$ for $c \neq d$. Then we can express the functional dependence of the ELBO on $\gamma_d$ as follows.

$$\mathrm{ELBO}(\gamma_d) = \sum_{k=1}^{K} (\alpha_k - 1)\left( \psi(\gamma_{dk}) - \psi\Big( \sum_{j=1}^{K} \gamma_{dj} \Big) \right) + \sum_{n=1}^{N_d} \sum_{k=1}^{K} \phi_{dw_{dn}k} \left( \psi(\gamma_{dk}) - \psi\Big( \sum_{j=1}^{K} \gamma_{dj} \Big) \right)$$

$$- \log \Gamma\left( \sum_{k=1}^{K} \gamma_{dk} \right) + \sum_{k=1}^{K} \log \Gamma(\gamma_{dk}) - \sum_{k=1}^{K} (\gamma_{dk} - 1)\left( \psi(\gamma_{dk}) - \psi\Big( \sum_{j=1}^{K} \gamma_{dj} \Big) \right)$$

$$+ \text{const}$$

$$= \sum_{k=1}^{K} \left( \alpha_k - \gamma_{dk} + \sum_{n=1}^{N_d} \phi_{dw_{dn}k} \right)\left( \psi(\gamma_{dk}) - \psi\Big( \sum_{j=1}^{K} \gamma_{dj} \Big) \right)$$

$$- \log \Gamma\left( \sum_{k=1}^{K} \gamma_{dk} \right) + \sum_{k=1}^{K} \log \Gamma(\gamma_{dk}) + \text{const}$$

The partial derivative of $\mathrm{ELBO}(\gamma_d)$ with respect to one of the dimensions of $\gamma_d$, say $\gamma_{dk}$, is

$$\frac{\partial}{\partial \gamma_{dk}} \mathrm{ELBO}(\gamma_d)$$

$$= -\left( \psi(\gamma_{dk}) - \psi\Big( \sum_{j=1}^{K} \gamma_{dj} \Big) \right) + \left( \alpha_k - \gamma_{dk} + \sum_{n=1}^{N_d} \phi_{dw_{dn}k} \right)\left( \psi_1(\gamma_{dk}) - \psi_1\Big( \sum_{j=1}^{K} \gamma_{dj} \Big) \right)$$

$$- \sum_{i:i \neq k} \left( \alpha_i - \gamma_{di} + \sum_{n=1}^{N_d} \phi_{dw_{dn}i} \right) \psi_1\Big( \sum_{j=1}^{K} \gamma_{dj} \Big) - \psi\Big( \sum_{j=1}^{K} \gamma_{dj} \Big) + \psi(\gamma_{dk})$$

$$= \psi_1(\gamma_{dk})\left( \alpha_k - \gamma_{dk} + \sum_{n=1}^{N_d} \phi_{dw_{dn}k} \right) - \psi_1\Big( \sum_{j=1}^{K} \gamma_{dj} \Big) \sum_{j=1}^{K} \left( \alpha_j - \gamma_{dj} + \sum_{n=1}^{N_d} \phi_{dw_{dn}j} \right).$$

As for the $\lambda$ case above, one obvious way to achieve a gradient of $\mathrm{ELBO}(\gamma_d)$ equal to zero is to set

$$\gamma_{dk} \leftarrow \alpha_k + \sum_{n=1}^{N_d} \phi_{dw_{dn}k} \quad \text{for } k = 1, \ldots, K.$$

Equivalently,

$$\gamma_{dk} \leftarrow \alpha_k + \sum_{v=1}^{V} n_{dv}\, \phi_{dvk} \quad \text{for } k = 1, \ldots, K.$$

**Variational parameter $\phi$.** Finally, consider fixing $\lambda$, $\gamma$, and $\phi_{cu}$ for $(c, u) \neq (d, v)$. In this case, the dependence of the ELBO on $\phi_{dv}$ can be written as follows.

$$\mathrm{ELBO}(\phi_{dv})$$

$$= \sum_{k=1}^{K} n_{dv} \, \phi_{dvk} \left( \psi(\gamma_{dk}) - \psi\Big( \sum_{j=1}^{K} \gamma_{dj} \Big) \right)$$

$$+ \sum_{k=1}^{K} n_{dv} \, \phi_{dvk} \left( \psi(\lambda_{kv}) - \psi\Big( \sum_{u=1}^{V} \lambda_{ku} \Big) \right) - \sum_{k=1}^{K} n_{dv} \, \phi_{dvk} \log \phi_{dvk} + \text{const}$$

$$= \sum_{k=1}^{K} n_{dv} \, \phi_{dvk} \left( - \log \phi_{dvk} + \psi(\gamma_{dk}) - \psi\Big( \sum_{j=1}^{K} \gamma_{dj} \Big) + \psi(\lambda_{kv}) - \psi\Big( \sum_{u=1}^{V} \lambda_{ku} \Big) \right)$$

$$+ \text{const}$$

The partial derivative of $\text{ELBO}(\phi_{dv})$ with respect to one of the dimensions of $\phi_{dv}$, say $\phi_{dvk}$, is

$$\frac{\partial}{\partial \phi_{dvk}} \text{ELBO}(\phi_{dv})$$

$$= n_{dv} \left( - \log \phi_{dvk} + \psi(\gamma_{dk}) - \psi\Big( \sum_{j=1}^{K} \gamma_{dj} \Big) + \psi(\lambda_{kv}) - \psi\Big( \sum_{u=1}^{V} \lambda_{ku} \Big) - 1 \right).$$

Using the method of Lagrange multipliers to incorporate the constraint that $\sum_{k=1}^{K} \phi_{dvk} = 1$, we wish to find $\rho$ and $\phi_{dvk}$ such that

$$0 = \frac{\partial}{\partial \phi_{dvk}} \left[ \text{ELBO}(\phi_{dv}) - \rho \left( \sum_{k=1}^{K} \phi_{dvk} - 1 \right) \right]. \tag{9}$$

Setting

$$\phi_{dvk} \propto_k \exp \left( \psi(\gamma_{dk}) - \psi\Big( \sum_{j=1}^{K} \gamma_{dj} \Big) + \psi(\lambda_{kv}) - \psi\Big( \sum_{u=1}^{V} \lambda_{ku} \Big) \right)$$

achieves the desired outcome in Eq. (9). Here, $\propto_k$ indicates that the proportionality is across $k$. The optimal choice of $\rho$ is expressed via this proportionality. The above assignment may also be written as

$$\phi_{dvk} \propto_k \exp \left( \mathbb{E}_q[\log \theta_{dk}] + \mathbb{E}_q[\log \beta_{kv}] \right)$$

The coordinate-ascent algorithm iteratively updates the parameters $\lambda$, $\gamma$, and $\phi$. In practice, we usually iterate the updates for the "local" parameters $\phi$ and $\gamma$ until they converge, then update the "global" parameter $\lambda$, and repeat. The resulting batch variational Bayes algorithm is presented in Alg. 1.

## A.2  SDA-Bayes VB

For a fixed hyperparameter $\alpha$, we can think of BatchVB as an algorithm that takes input in the form of a prior on topic parameters $\beta$ and a minibatch of documents. In particular, let $C_b$ be the $b$th minibatch of documents; for documents with indices in $\mathcal{D}_b$, these documents can be summarized by the word counts $(n_d)_{d \in \mathcal{D}_b}$. Then, in the notation of Eq. (2), we have $\Theta = \beta$, $\mathcal{A} = \text{BatchVB}$, and

$$q_0(\beta) = \prod_{k=1}^{K} \text{Dirichlet}(\beta_k | \eta_k).$$

In general, the $b$th posterior takes the same form and therefore can be summarized by its parameters $\lambda^{(b)}$:

$$q_b(\beta) = \prod_{k=1}^{K} \text{Dirichlet}(\beta_k | \lambda_k^{(b)}).$$

In this case, if we set the prior parameters to $\lambda_k^{(0)} := \eta_k$, Eq. (2) becomes the following algorithm.

---

**Algorithm 4:** Streaming VB for LDA

---

**Input**: Hyperparameter $\eta$
Initialize $\lambda^{(0)} \leftarrow \eta$
**foreach** *Minibatch $C_b$ of documents* **do**
$\quad \lambda^{(b)} \leftarrow \text{BatchVB}\left(C_b, \lambda^{(b-1)}\right)$
$\quad q_b(\beta) = \prod_{k=1}^{K} \text{Dirichlet}(\beta_k | \lambda_k^{(b)})$

---

Next, we apply the asynchronous, distributed updates described in the "Asynchronous Bayesian updating" portion of Sec. 2 to the batch VB primitive and LDA model. In this case, $\lambda^{(\text{post})}$ is the posterior parameter estimate maintained at the master, and each worker updates this value after a local computation. The posterior after seeing a collection of minibatches is $q(\beta) = \prod_{k=1}^{K} \text{Dirichlet}(\beta_k | \lambda_k^{(\text{post})})$.

---

**Algorithm 5:** SDA-Bayes with VB primitive for LDA

---

**Input**: Hyperparameter $\eta$
Initialize $\lambda^{(\text{post})} \leftarrow \eta$
**foreach** *Minibatch $C_b$ of documents, at a worker* **do**
$\quad$ Copy master value locally: $\lambda^{(local)} \leftarrow \lambda^{(\text{post})} \ \lambda \leftarrow \text{BatchVB}\left(C_b, \lambda^{(\text{local})}\right)$
$\quad \Delta\lambda \leftarrow \lambda - \lambda^{(\text{local})}$
$\quad$ Update the master value synchronously: $\lambda^{(\text{post})} \leftarrow \lambda^{(\text{post})} + \Delta\lambda$

---

# B  Expectation Propagation

## B.1  Batch EP

Our batch expectation propagation (EP) algorithm for LDA learns a posterior for both the document-specific topic mixing proportions $(\theta_d)_{d=1}^{D}$ and the topic distributions over words $(\beta_k)_{k=1}^{K}$. By contrast, the algorithm in [14] learns only the former and so is not appropriate to the model in Sec. 3.

For consistency, we also follow [14] in making a distinction between token and type word updates, where a token refers to a particular word instance and a type refers to all words with the same vocabulary value. Let $C = (w_d)_{d=1}^{D}$ denote the set of documents that we observe, and for each word $v$ in the vocabulary, let $n_{dv}$ denote the number of times $v$ appears in document $d$.

**Collapsed posterior.** We begin by collapsing (i.e., integrating out) the word assignments $z$ in the posterior (7) of LDA. We can express the collapsed posterior as

$$p(\beta, \theta \mid C, \eta, \alpha) \propto \left[\prod_{k=1}^{K} \text{Dirichlet}_V(\beta_k \mid \eta_k)\right] \cdot \prod_{d=1}^{D} \left[\text{Dirichlet}_K(\theta_d \mid \alpha) \cdot \prod_{v=1}^{V} \left(\sum_{k=1}^{K} \theta_{dk}\, \beta_{kv}\right)^{n_{dv}}\right].$$

For each document-word pair $(d, v)$, consider approximating the term $\sum_{k=1}^{K} \theta_{dk}\beta_{kv}$ above by

$$\left[\prod_{k=1}^{K} \text{Dirichlet}_V(\beta_k \mid \chi_{kdv} + \mathbf{1}_V)\right] \cdot \text{Dirichlet}_K(\theta_d \mid \zeta_{dv} + \mathbf{1}_K),$$

where $\chi_{kdv} \in \mathbb{R}^V$, $\zeta_{dv} \in \mathbb{R}^K$, and $\mathbf{1}_M$ is a vector of all ones of length $M$. This proposal serves as inspiration for taking the approximating variational distribution for $p(\beta, \theta \mid C, \eta, \alpha)$ to be of the form

$$q(\beta, \theta \mid \lambda, \gamma) := \left[\prod_{k=1}^{K} q(\beta_k \mid \lambda_k)\right] \cdot \prod_{d=1}^{D} q(\theta_d \mid \gamma_d), \tag{10}$$

where $q(\beta_k \mid \lambda_k) = \text{Dirichlet}(\beta_k \mid \lambda_k)$ and $q(\theta_d \mid \gamma_d) = \text{Dirichlet}(\theta_d \mid \gamma_d)$, with the parameters

$$\lambda_k = \eta_k + \sum_{d=1}^{D} \sum_{v=1}^{V} n_{dv} \chi_{kdv}, \qquad \gamma_d = \alpha + \sum_{v=1}^{V} n_{dv} \zeta_{dv}, \tag{11}$$

and the constraints $\lambda_k \in \mathbb{R}_+^V$ and $\gamma_d \in \mathbb{R}_+^K$ for each $k$ and $d$. We assume this form in the remainder of the analysis and write $q(\beta, \theta \mid \chi, \zeta)$ for $q(\beta, \theta \mid \lambda, \gamma)$, where $\chi = (\chi_{kdv})$, $\zeta = (\zeta_{dv})$.

**Optimization problem.** We seek to find the optimal parameters $(\chi, \zeta)$ by minimizing the (reverse) KL divergence:

$$\min_{\chi, \zeta} \ \mathrm{KL}\left(p(\beta, \theta \mid C, \eta, \alpha) \,\|\, q(\beta, \theta \mid \chi, \zeta)\right).$$

This joint minimization problem is not tractable, and the idea of EP is to proceed iteratively by fixing most of the factors in Eq. (10) and minimizing the KL divergence over the parameters related to a single word.

More formally, suppose we already have a set of parameters $(\chi, \zeta)$. Consider a document $d$ and word $v$ that occurs in document $d$ (i.e., $n_{dv} \geq 1$). We start by removing the component of $q$ related to $(d, v)$ in Eq. (10). Following [7], we subtract out the effect of one occurrence of word $v$ in document $d$, but at the end of this process we update the distribution on the type level. In doing so, we use the following shorthand for the remaining global parameters:

$$\lambda_k^{\backslash(d,v)} = \lambda_k - \chi_{kdv} = \eta_k + (n_{dv} - 1)\chi_{kdv} + \sum_{(d',v'):(d',v')\neq(d,v)} n_{d'v'}\chi_{kd'v'}$$

$$\gamma_d^{\backslash(d,v)} = \gamma_d - \zeta_{dv} = \alpha + (n_{dv} - 1)\zeta_{dv} + \sum_{v':v'\neq v} n_{dv'}\zeta_{dv'}.$$

We replace this removed part of $q$ by the term $\sum_{k=1}^{K} \theta_{dk}\beta_{kv}$, which corresponds to the contribution of one occurrence of word $v$ in document $d$ to the true posterior $p$. Call the resulting normalized distribution $\tilde{q}_{dv}$, so $\tilde{q}_{dv}(\beta, \theta \mid \lambda^{\backslash(d,v)}, \gamma_{\backslash d}, \gamma_d^{\backslash(d,v)})$ satisfies

$$\propto \left[\prod_{k=1}^{K} \mathrm{Dirichlet}(\beta_k \mid \lambda_k^{\backslash(d,v)})\right] \cdot \left[\prod_{d'\neq d} \mathrm{Dirichlet}(\theta_{d'} \mid \gamma_{d'})\right] \cdot \mathrm{Dirichlet}(\theta_d \mid \gamma_d^{\backslash(d,v)}) \cdot \sum_{k=1}^{K} \theta_{dk}\,\beta_{kv}.$$

We obtain an improved estimate of the posterior $q$ by updating the parameters from $(\lambda, \gamma)$ to $(\hat{\lambda}, \hat{\gamma})$, where

$$(\hat{\lambda}, \hat{\gamma}) = \arg\min_{\lambda', \gamma'} \ \mathrm{KL}\left(\tilde{q}_{dv}(\beta, \theta \mid \lambda^{\backslash(d,v)}, \gamma_{\backslash d}, \gamma_d^{\backslash(d,v)}) \,\|\, q(\beta, \theta \mid \lambda', \gamma')\right). \qquad (12)$$

**Solution to the optimization problem.** First, note that for $d' : d' \neq d$, we have $\hat{\gamma}_{d'} = \gamma_{d'}$.

Now consider the index $d$ chosen on this iteration. Since $\beta$ and $\theta$ are Dirichlet-distributed under $q$, the minimization problem in Eq. (12) reduces to solving the moment-matching equations [7, 20]

$$\mathbb{E}_{\tilde{q}_{dv}}[\log \beta_{ku}] = \mathbb{E}_{\hat{\lambda}_k}[\log \beta_{ku}] \qquad \text{for } 1 \leq k \leq K, \ 1 \leq u \leq V,$$
$$\mathbb{E}_{\tilde{q}_{dv}}[\log \theta_{dk}] = \mathbb{E}_{\hat{\gamma}_d}[\log \theta_{dk}] \qquad \text{for } 1 \leq k \leq K.$$

These can be solved via Newton's method though [7] recommends solving exactly for the first and "average second" moments of $\beta_{ku}$ and $\theta_{dk}$, respectively, instead. We choose the latter approach for consistency with [7]; our own experiments also suggested taking the approach of [7] was faster than Newton's method with no noticeable performance loss. The resulting moment updates are

$$\hat{\lambda}_{ku} = \frac{\sum_{y=1}^{V} \left(\mathbb{E}_{\tilde{q}_{dv}}[\beta_{ky}^2] - \mathbb{E}_{\tilde{q}_{dv}}[\beta_{ky}]\right)}{\sum_{y=1}^{V} \left(\mathbb{E}_{\tilde{q}_{dv}}[\beta_{ky}]^2 - \mathbb{E}_{\tilde{q}_{dv}}[\beta_{ky}^2]\right)} \cdot \mathbb{E}_{\tilde{q}_{dv}}[\beta_{ku}] \qquad (13)$$

$$\hat{\gamma}_{dk} = \frac{\sum_{j=1}^{K} \left(\mathbb{E}_{\tilde{q}_{dv}}[\theta_{dj}^2] - \mathbb{E}_{\tilde{q}_{d,n}}[\theta_{dj}]\right)}{\sum_{j=1}^{K} \left(\mathbb{E}_{\tilde{q}_{dv}}[\theta_{dj}]^2 - \mathbb{E}_{\tilde{q}_{dv}}[\theta_{dj}^2]\right)} \cdot \mathbb{E}_{\tilde{q}_{dv}}[\theta_{dk}]. \qquad (14)$$

We then set $(\chi_{kdv})_{k=1}^K$ and $\zeta_{dv}$ such that the new global parameters $(\lambda_k)_{k=1}^K$ and $\gamma_d$ are equal to the optimal parameters $(\hat\lambda_k)_{k=1}^K$ and $\hat\gamma_d$. The resulting algorithm is presented below (Alg. 6).

---

**Algorithm 6:** EP for LDA

---

**Input**: Data $C = (w_d)_{d=1}^D$; hyperparameters $\eta, \alpha$
**Output**: $\lambda$
Initialize $\forall (k, d, v)$, $\chi_{kdv} \leftarrow 0$ and $\zeta_{dv} \leftarrow 0$
**while** $(\chi, \zeta)$ *not converged* **do**

    **foreach** $(d, v)$ *with* $n_{dv} \geq 1$ **do**

        `/* Variational distribution without the word token` $(d,v)$    `*/`

        $\forall k, \; \lambda_k^{\backslash(d,v)} \leftarrow \eta_k + (n_{dv} - 1)\chi_{kdv} + \sum_{(d',v') \neq (d,v)} n_{d'v'} \chi_{kd'v'}$

        $\gamma_d^{\backslash(d,v)} \leftarrow \alpha + (n_{dv} - 1)\zeta_{dv} + \sum_{v' \neq v} n_{dv'} \zeta_{dv'}$

        If any of $\lambda_{ku}^{\backslash(d,v)}$ or $\gamma_{dk}^{\backslash(d,v)}$ are non-positive, skip updating this $(d, v)$      (†)

        `/* Variational parameters from moment-matching`       `*/`

        $\forall (k, u)$, compute $\hat\lambda_{ku}$ from Eq. (13)

        $\forall k$, compute $\hat\gamma_{dk}$ from Eq. (14)

        `/* Type-level updates to parameter values`        `*/`

        $\forall k, \; \chi_{kdv} \leftarrow n_{dv}^{-1} \left( \hat\lambda_k - \lambda_k^{\backslash(d,v)} \right) + \left( 1 - n_{dv}^{-1} \right) \chi_{kdv}$

        $\zeta_{dv} \leftarrow n_{dv}^{-1} \left( \hat\gamma_d - \gamma_d^{\backslash(d,v)} \right) + \left( 1 - n_{dv}^{-1} \right) \zeta_{dv}$

        Other $\chi, \zeta$ remain unchanged

`/* Global variational parameters`               `*/`
$\forall k, \; \lambda_k \leftarrow \eta_k + \sum_{d=1}^D \sum_{v=1}^V n_{dv} \chi_{kdv}$

---

The results in the main text (Sec. 4) are reported for Alg. 6. We also tried a slightly modified EP algorithm that makes token-level updates to parameter values, rather than type-level updates. This modified version iterates through each word *placeholder* in document $d$; that is, through pairs $(d, n)$ rather than pairs $(d, v)$ corresponding to word *values*. Since there are always at least as many $(d, n)$ pairs as $(d, v)$ pairs with $n_{dv} \geq 1$ (and usually many more of the former), the modified algorithm requires many more iterations. In practice, we find better experimental performance for the modified EP algorithm in terms of log predictive probability as a function of number of data points in the training set seen so far: e.g., leveling off at about $-7.96$ for Nature vs. $-8.02$. However, the modified algorithm is also much slower, and still returns much worse results than SDA-Bayes or SVI, so we do not report these results in the main text.[3]

## B.2 SDA-Bayes EP

Putting a batch EP algorithm for LDA into the SDA-Bayes framework is almost identical to putting a batch VB algorithm for LDA into the SDA-Bayes framework. This similarity is to be expected since SDA-Bayes works out of the box with a batch approximation algorithm in the correct form.

For a fixed hyperparameter $\alpha$, we can think of BatchEP as an algorithm (just like BatchVB) that takes input in the form of a prior on topic parameters $\beta$ and a minibatch of documents. The same

setup and notation from Sup. Mat. A.2 applies. In this case, Eq. (2) becomes the following algorithm.

---

**Algorithm 7:** Streaming EP for LDA

---

**Input**: Hyperparameter $\eta$
Initialize $\lambda^{(0)} \leftarrow \eta$
**foreach** *Minibatch $C_b$ of documents* **do**
$\quad \mid \quad \lambda^{(b)} \leftarrow \text{BatchEP}\left(C_b, \lambda^{(b-1)}\right)$
$\quad \mid \quad q_b(\beta) = \prod_{k=1}^{K} \text{Dirichlet}(\beta_k | \lambda_k^{(b)})$

---

This algorithm is exactly the same as Alg. 4 but with a batch EP primitive instead of a batch VB primitive.

Next, we apply the asynchronous, distributed updates described in the "Asynchronous Bayesian updating" portion of Sec. 2 to the batch EP primitive and LDA model. Again, the setup and notation from Sup. Mat. A.2 applies, and we find the following algorithm.

---

**Algorithm 8:** SDA-Bayes with EP primitive for LDA

---

**Input**: Hyperparameter $\eta$
Initialize $\lambda^{(\text{post})} \leftarrow \eta$
**foreach** *Minibatch $C_b$ of documents, at a worker* **do**
$\quad \mid \quad$ Copy master value locally: $\lambda^{(local)} \leftarrow \lambda^{(\text{post})} \; \lambda \leftarrow \text{BatchEP}\left(C_b, \lambda^{(\text{local})}\right)$
$\quad \mid \quad \Delta\lambda \leftarrow \lambda - \lambda^{(\text{local})}$
$\quad \mid \quad$ Update the master value synchronously: $\lambda^{(\text{post})} \leftarrow \lambda^{(\text{post})} + \Delta\lambda$

---

Indeed, the recipe outlined here applies more generally to other primitives besides EP and VB.

## Footnotes

[3]Here and in the main text we run EP with $\eta = 1$. We also tried EP with $\eta = 0.01$, but the positivity check for $\lambda_{ku}^{\backslash(d,v)}$ and $\gamma_{dk}^{\backslash(d,v)}$ on line (†) in Algorithm 6 always failed and as a result none of the parameters were updated.