[Reviews · NeurIPS 2013]

Submitted by Assigned_Reviewer_4

The paper presents a streaming algorithm for inference in topic models. After each mini-batch, the algorithm maintains the posterior over the model parameter given the data seen thus far. This in contrast to variational LDA where the quantity maintained is an approximation to the posterior of the full data set (which require knowing apriori the size of the dataset). The resulting algorithm can be used in truly streaming settings, doesn't require setting learning rates and is less sensitive to mini-batch size.

Overall this is an interesting idea. Few comments:

1) regarding evaluation, I am a bit concerned that you *only* ran with a Dirichlet parameter of 1 (line 312) over \beta (topic word distribution). Usually values of .01, .1 or smaller are used for this hyper-parameter to encourage sparse topics. In the SVI paper, a value of .01 was used. I am really curious to see how things would change as you vary that prior. Wouldn't that initial prior matter more in your case since the batch size is more or less "given" thus the rate at which this prior is overwritten is controlled by that parameter. In SVI the learning rate would help to some degree. Some analysis is required here.

2) I think the main contribution of the paper is the departure from approximating the final posterior to approximating the "running posterior".

3) You can still throw many threads at SVI and scale it (just distribute the VI step over the documents in the mini-batch), why not? Please update table 1 based on that.

4) You can still use Hogwild style update over the natural gradient in SVI as well. Why not?

With that in mind I suggest framing the main contribution around 1 (which is great by itself) unless there is a good argument against 2 and 3 in SVI.

5) nit. You only ran the algorithm in multi-threaded mode. Hogwild itself was proposed as a multi-threaded algorithm and getting hogwild to run in truly distributed settings (across machines) is not trivial with network delay and what not. Please discuss this in the paper.

6) It would help to see how the topics evolve as documents arrive akin to the SVI paper.


Summary: A streaming inference algorithm for LDA using variational Bayes recursive updates. Less sensitive to tuning parameters than existing approaches (SVI) and gives comparable results. Few points in the evaluation need to be discussed/considered to make sure that this result carry over to other hyper-parameter settings.

Submitted by Assigned_Reviewer_5

AC:

This paper presents an interesting idea. It is very well-written and
well-executed. It is a clear accept.

Authors:

This was a fascinating paper. Thank you. I have some comments and
criticisms. I would appreciate a response in the author rebuttal.

I agree with the main idea around your paper: SVI cannot really be
used in a true streaming setting because we have to set D, the data
set size. (Note: In the final version of our long paper about SVI, we
removed references to "streaming data" and cited this as a point for
future work.) Further issues with SVI are its sensitivity to the
learning rate and mini-batch size. Your taking the perspective of
incremental Bayesian updating, as an alternative to SVI, is very
elegant.

- You frame SVI as being "designed" for the single pass setting (e.g.,
in the abstract, p5, and elsewhere). This isn't our perspective about
it. Rather, SVI repeatedly subsamples mini-batches, with replacement,
from the data set. In many applications, a user may want to treat a
single pass, divided into mini-batches, as though this were the case.
But note that this is different from the original conception of the
algorithm. I think you should make this clear.

- If we were to use SVI in a streaming setting and had to artificially
set D, this would essentially interact with setting the learning rate
parameters. I suggest mentioning this in your paper.

- The parallelized asynchronous component is very nice, but the reader
might mistakenly think here that it only applies to SDA-Bayes. You
should note, though it is not the subject of your paper, that this
innovation applies as easily to SVI. As with SDA-Bayes, one can march
forward with asynchronous updates (despite the lack of theory) and
speed up the algorithm.

- My reading of the results is this. SDA-Bayes performs as well as SVI
if we run SVI with one pass through the data. It comes from a
different perspective about approximating the posterior, does not
require setting a learning rate, and handles infinite sets of data
without having to make up a data set size.

- That said, I'm not sure the comparison to the multi-threaded version
is fair. Doesn't that version effectively set the minibatch size to
8,192 (32 * 256)? With that minibatch size, how do the algorithms
compare? (I emphasize that I don't think it matters: getting around
learning rates and D is a significant contribution.)

- In showing how SVI is sensitive to learning rate and D, I suggest
putting SDA-Bayes (at the same mini-batch size) as a horizontal line.
This will emphasize that while users must explore these parameters in
SVI, they don't need to with your algorithm.

Small comments:

- p4: "utilized" -> "used" (optional, of course)

- p6: does [3] use a subset? It does to compare to batch, but I
believe we also analyzed the full Wikipedia corpus.

Citation to mention:

- Ranganath et al. have worked on automatically setting learning rates
in SVI:

R. Ranganath, C. Wang, D. Blei, and E. Xing. An adaptive learning rate
for stochastic variational inference. International Conference on
Machine Learning, 2013
Summary: This paper presents an interesting idea. It is very well-written and
well-executed. It is a clear accept.

Submitted by Assigned_Reviewer_6

The authors present a Streaming Distributed Asynchronous (SDA) framework for variational Bayesian computation aimed at analysis of large datasets. They assume a variational form q(θ) for the posterior on model parameters that is exponential family, which they express as q(θ) ∝ exp[ξ . T(θ)]. Note that this parameter vector ξ = (χ ν) contains both the cumulant of the conjugate sufficient statistics χ and the scale parameter ν. The authors then define a set of parameters ξ_0 for the variational approximation of the prior p(θ), and make the observation that when the data is split up into batches, independent posteriors ξ_b for each batch may be combined to obtain a approximation for the posterior on the aggregate data ξ = ξ_0 + Σ_b (ξ_b - ξ_0). Τhey propose an algorithm where each new batch is used to update ξ_0, i.e. for each new batch prior is taken to be the posterior after observation of the previous batch. Τhis algorithm can be run in a distributed setting where each worker returns Δξ_b = (ξ_b - ξ_0) to the master, which then updates its copy of ξ_0 <- ξ_0 + Δξ_b accordingly.

Τhe proposed methodology is simple but useful, primarily because it presents a trivial parallelization strategy that may be applied to any algorithm that obtains an exponential family estimate of a posterior p(θ | x). The main competitor to this method is stochastic variational inference (SVI), a method that uses a stochastic optimization based on a gradient estimate of the lower bound evidence. A disadvantage of SVI, as the authors note, is that it is sensitive to parameters for the total number of expected documents, batch size, and step size. However I am not entirely convinced this is always a bad thing. While tuning parameters can indeed be costly, having such tunable parameters also makes a method more flexible. The presented results on LDA are promising, but they are not sufficient to demonstrate that SDA can match or exceed SVI results in a variety of domains while eliminating the need for such tunable settings.

I would have also liked to have seen a clearer motivation for the proposed distributed update scheme. When comparing their chosen update scheme with one where ξ_0 is held constant for each of the workers, the authors note that the "key difference between the first and second frameworks proposed above is that, in the second, the latest posterior is used as a prior. This latter framework is more in line with the streaming update of Eq. (2) but introduces a new layer of approximation". Unfortunately the implications of this approximation are left unexplored, save for the otherwise unqualified statement that "we find that the latter framework performs better in practice, so we focus on it exclusively in what follows".

It would seem that the essential difference between SDA and SVI is that in SDA ξ grows with the number of documents, whereas the variational parameters have a fixed strength in SVI. This implies that in SDA (1) the entropy of the posterior q(θ) decreases as more data is seen, and that (2) convergence is ensured by virtue of the fact that in later updates (ξ_b - ξ_0) << ξ_0. In SVI convergence is achieved by annealing the step size to 0 and ξ has a fixed strength determined by the total number of documents D, which in an on-line setting one might interpret as a buffer size. It would be interesting to see a more detailed analysis of which of these two strategies is more effective in terms of ensuring rapid convergence while avoiding local maxima.

In short I believe this may be a promising approach to variational inference in large data sets, chiefly because it is by design suitable to distributed inference in a map-reduce type setting. However I do feel more thorough testing is needed before this method can be considered a drop-in replacement for SVI.

--
Minor points / Questions

• I see no reason why SVI could not be performed with distributed updates precisely in the manner proposed here for SDA. Have the authors considered this scenario?

• Are the authors using their own implementation of SVI in the analysis comparison, Matt Hoffman's code (http://www.cs.princeton.edu/~mdhoffma/code/onlineldavb.tar) or the Vowpal Wabbit implementation (http://hunch.net/~vw)? Given that a run-time comparison is performed, it would be desirable to state this explicitly

• The authors write that in SVI "the stochastic gradient is computed for a single data point (document) at a time". This statement is technically correct but somewhat confusing, since it suggests SVI is performed on single data-points, not batches.
Summary: This paper presents a simple and interesting approach to variational inference in large datasets that is by design suitable to a distributed map-reduce type setting, and at the same time eliminates the step size parameters that need to be tuned in SVI. The experimental results however, which only consider LDA, leave some question as to the comparative strengths and weaknesses relative to SVI.

Submitted by Assigned_Reviewer_7

The paper proposes an inference framework (SDA) for streaming data in the Bayesian setting. This framework exploits the additive relationship between new data points and exponential family natural parameters, yielding a Bayesian streaming/online update equation that (a) does not require prior knowledge of the number of data points (unlike SVI), (b) has fewer tuning parameters than SVI, and (c) in the case of exact update equations, is perfectly suited to asynchronous updates (though in practice, approximate update equations are used, making the effect of asynchrony somewhat unclear). While SDA single-threaded performance is about 4 times slower than SVI, SDA's lack of a need to tune parameters and good asynchronous performance are promising for large-scale distributed inference, particularly at large cluster sizes.

Quality:

The arguments and derivations appear sound, while the experimental evidence is adequate (though a wider variety of datasets would have been welcome, such as the NY Times dataset). However, I found the description of how SDA was applied to LDA using VB to be insufficient. As a matter of completeness, the authors should have provided an explicit, algorithmic description of how SDA uses VB to update the LDA natural parameters, just as they did for EP in the appendix.

Minor comment: In section 2.3, the authors claim that the "second framework works better in practice", despite the fact it introduces additional approximation. I wonder if this is simply a consequence of using VB, which is a kind of local approximation? It seems that giving the workers a recent (albeit inaccurate) version of the posterior parameters should yield a better VB approximation than the prior parameters.

Clarity:

I found the writing to be somewhat lacking in intuition, and a little difficult to follow. I felt as if the key ideas were lost in the math and descriptions, and I had to read multiple passes before I could appreciate the value of SDA.

Originality:

While the idea of updating natural family parameters is not new within statistics, its application to streaming data on complex models that require approximating distributions is new to me. I would say the paper is original in the context of large-scale machine learning on complex models.

Significance:

I believe SDA provides significant utility for large-scale ML, despite being slower than SVI currently. SDA is closer to "working out of the box" than SVI, because SDA has no tuning parameters and does not depend on the number of documents. The fact that SDA can be run asynchronously is a big plus at large scales.
Summary: SDA provides significant usability advantages over SVI, though it is currently slower. Some technical and clarity issues detract from the paper's quality, but I nevertheless found it valuable.
Author Feedback

Author rebuttal: We are very grateful to the reviewers for their insightful comments, helpful suggestions, and errors spotted. On a general note, we agree with the reviewers that our main goal was to demonstrate a method for calculating the "running posterior" as opposed to a single posterior. We meant for the distributed, asynchronous part to show that our algorithm can be run quickly and robustly in practice, not to differentiate it from SVI. We will endeavor to make this distinction more clear throughout the text. We agree that SVI's computations may be distributed asynchronously, as for general stochastic gradient descent. We respond to other comments in more detail below.

Reviewer_4

1) I am a bit concerned that you *only* ran with a Dirichlet parameter of 1 (line 312) over \beta (topic word distribution). Usually values of .01, .1 or smaller are used for this hyper-parameter

Thank you for noticing this discrepancy. We will examine more carefully how the value of eta affects both SDA-Bayes and SVI in further experiments.

5) nit. You only ran the algorithm in multi-threaded mode. Hogwild itself was proposed as a multi-threaded algorithm and getting hogwild to run in truly distributed settings (across machines) is not trivial with network delay and what not. Please discuss this in the paper.

It is fair to note we use multiple processors rather than multiple machines. However, we feel that discussing this in more depth would be somewhat far from the main thrust of the paper (the running approximation). We do hope to work in distributed settings across machines in the future, particularly given that our algorithm is naturally suited to a map-reduce implementation.

6) It would help to see how the topics evolve as documents arrive akin to the SVI paper.

This would be interesting for future work (in particular, if the dataset is changing over time), but our focus in this paper was to demonstrate an inference algorithm and not really to delve too far into topic models.

Reviewer_5

- You frame SVI as being "designed" for the single pass setting (e.g., in the abstract, p5, and elsewhere).

Thank you for pointing out this mistake; we agree and will correct our wording.

- That said, I'm not sure the comparison to the multi-threaded version is fair. Doesn't that version effectively set the minibatch size to 8,192 (32 * 256)? With that minibatch size, how do the algorithms compare?

This is true. Our main goal with the multi-threaded version was to show that SDA-Bayes can be run in a reasonable amount of time (which we believe is key to practical streaming). For minibatch size, all of our lines in Figure 3 are for the single-thread case to facilitate direct comparison between SVI and SDA across this dimension.

- p6: does [3] use a subset? It does to compare to batch, but I believe we also analyzed the full Wikipedia corpus.

Thank you for calling this error in our discussion to our attention.

Reviewer_6

• The presented results on LDA are promising, but they are not sufficient to demonstrate that SDA can match or exceed SVI results in a variety of domains while eliminating the need for such tunable settings.

We disagree that eliminating the SVI tuning parameters is the main contribution of this paper. Rather, we see this work as solving fundamentally different problems from SVI: finding a running posterior, working out of the box for (any) model that already has a batch approximation in the same exponential family as the prior, and doing these things in reasonable time.

• I would have also liked to have seen a clearer motivation for the proposed distributed update scheme.

We hope to examine this interesting phenomenon more carefully in future work. We hypothesize that this effect may occur due to the lack of identifiability between different permutations of topics. Using the latest posterior for each minibatch calculation may help focus down a particular permutation rather than averaging over all permutations. But this is merely conjecture.

• It would be interesting to see a more detailed analysis of which of these two strategies is more effective in terms of ensuring rapid convergence while avoiding local maxima.

We believe that convergence in SVI is fundamentally different from inference in SDA-Bayes. In the SVI case, D' data points (with D' << D) might be sufficient to quickly estimate the posterior for D data points. In the SDA-Bayes case, the number of data points processed corresponds to the size of the posterior being estimated.

• Are the authors using their own implementation of SVI in the analysis comparison

On line 310, we note that we use Matt Hoffman's code and give a link in citation [17].

• The authors write that in SVI "the stochastic gradient is computed for a single data point (document) at a time". This statement is technically correct but somewhat confusing, since it suggests SVI is performed on single data-points, not batches.

We will correct this misleading wording.

• The experimental results however, which only consider LDA, leave some question as to the comparative strengths and weaknesses relative to SVI.

We preferred to study a single model in depth rather than say little about many models given the constrained space. We do hope to address other interesting, complex models in future work. We also note that the original SVI paper focused exclusively on LDA, a model of significant current interest.

Reviewer_7

- As a matter of completeness, the authors should have provided an explicit, algorithmic description of how SDA uses VB to update the LDA natural parameters, just as they did for EP in the appendix.

Excellent idea. We are happy to provide more detail in an additional appendix for our derivations.

- Minor comment: In section 2.3, the authors claim that the "second framework works better in practice", despite the fact it introduces additional approximation.

Please see our response to Reviewer_6.